# Genome-Wide Analysis of Dental Caries Variability Reveals Genotype-by-Environment Interactions

**DOI:** 10.3390/genes14030736

**Published:** 2023-03-17

**Authors:** Tianyu Zou, Betsy Foxman, Daniel W. McNeil, Seth M. Weinberg, Mary L. Marazita, John R. Shaffer

**Affiliations:** 1Department of Human Genetics, School of Public Health, University of Pittsburgh, Pittsburgh, PA 15261, USA; tiz45@pitt.edu (T.Z.);; 2Department of Epidemiology, University of Michigan Medical School, Ann Arbor, MI 48109, USA; 3Department of Community Dentistry and Behavioral Science, College of Dentistry, University of Florida, Gainesville, FL 32610, USA; 4Center for Craniofacial and Dental Genetics, Department of Oral and Craniofacial Sciences, School of Dental Medicine, University of Pittsburgh, Pittsburgh, PA 15219, USA; 5Clinical and Translational Sciences, School of Medicine, University of Pittsburgh, Pittsburgh, PA 15213, USA

**Keywords:** genome-wide vQTL, genotype-by-environment interactions, dental caries

## Abstract

Genotype-by-environment interactions (GEI) may influence dental caries, although their effects are difficult to detect. Variance quantitative trait loci (vQTL) may serve as an indicator of underlying GEI effects. The aim of this study was to investigate GEI effects on dental caries by prioritizing variants from genome-wide vQTL analysis. First, we identified vQTLs from ~4.3 M genome-wide variants in three cohorts of white children aged 3–5 (n = 396, n = 328, n = 773) using Levene’s test. A total of 39 independent vQTLs with *p* < 1 × 10^−6^ were identified, some of which were located in or near genes with plausible biological roles in dental caries (*IGFBP7*, *SLC5A8*, and *SHH* involved in tooth development and enamel mineralization). Next, we used linear regression to test GEI effects on dental caries with the 39 prioritized variants and self-reported environmental factors (demographic, socioeconomic, behavioral, and dietary factors) in the three cohorts separately. We identified eight significant GEIs indicating that children with vQTL risk genotypes had higher caries experience if they had less educated parents, lower household/parental income, brushed their teeth less frequently, consumed sugar-sweetened beverages more frequently, were not breastfed, and were female. We reported the first genome-wide vQTL analysis of dental caries in children nominating several novel genes and GEI for further investigations.

## 1. Introduction

The etiology of dental caries is complex, involving both genetic and environmental/behavioral risk and protective factors (e.g., diet, oral hygiene/behaviors, and fluoride exposure). The effects of genetic factors on caries risk may differ among individuals across environmental/social strata due to genotype-by-environment interactions (GEI). For individuals, the consequence of this complexity is that the impact of environmental risk factors on their susceptibility to caries varies with their genetics. Therefore, investigating GEI in dental caries can aid in discovering additional susceptibility loci for dental caries, identifying populations at high risk, and developing more accurate genetic risk prediction tools. However, GEI are challenging to study because they require relatively large samples with well-characterized phenotypes, genotypes, and environmental risk factors.

To date, few studies have investigated GEI for dental caries. Using candidate gene approaches, Shaffer et al. [1] tested the interaction effects between four enamel matrix genes with fluoride exposure on caries experience. They found participants with risk alleles of two genetic variants (upstream of *TUFT1* and missense in *AMBN*) showed higher levels of dental caries, but only if they lacked sufficient exposure to fluoride. Also using a candidate gene approach, Yildiz et al. [2] found weak evidence that the interactions between three genetic variants in genes *CA6*, *DEFB1*, and *TAS2R38* with dental plaque, lactobacilli count, age, and saliva buffer capacity associated with caries experience in 154 adults. A genome-wide GEI study of 709 US children evaluated the interaction between the presence of oral Streptococcus mutans, a putative caries-promoting bacteria, and host caries susceptibility [3]. Three suggestive loci in genes *IL32*, *GALK2*, and *CELF4* were identified that interacted with S. mutans, but no genome-wide significant signals were observed. The limited success of these prior studies may be due to the narrow range of genes and environmental factors tested and/or low power due to small sample sizes.

There are several well-known challenges that impede our ability to detect GEI effects in human studies. First, an accurate record of environmental risk factors is challenging to collect. Second, identifying GEI effects requires larger sample sizes compared to detecting main effects due to the potentially small effect sizes of GEI and increased dimensionality (i.e., joint genetic-environment strata) across which the effects are modeled. Moreover, these challenges are exacerbated in the context of genome-wide association studies (GWAS), where millions of SNPs, and potentially a large number of environments, are tested for GEI effects. A more efficient strategy is therefore needed to address these problems.

One such strategy is to target genetic variants known as variance quantitative trait loci (vQTL), which have effects on the variability of a trait [4], i.e., showing genetic variance heterogeneity. vQTL analysis provides a method to reduce the scope of GEI testing by prioritizing SNPs mostly likely to be involved in interactions and can be performed even in the absence of data on the interacting risk factor. The idea is that if a genetic variant is associated with the variance of a quantitative phenotype (i.e., the variant is a vQTL), then it may influence the phenotype by interacting with an environmental factor (i.e., GEI). Thus, a two-step approach can be utilized to preserve power for GEI analysis: (1) performing a genome-wide scan for variants with vQTL effects on the phenotype; (2) using the identified vQTLs in interaction modeling to detect GEI associated with the phenotype [5,6].

In this study, we applied the aforementioned two-stage design for detecting GEI in dental caries in several well-charactered cohorts of children. Specifically, we first performed genome-wide vQTL scans (in each cohort separately and combined via meta-analysis) and then tested prioritized discovered variants for interaction effects with several environmental risk factors.

## 2. Materials and Methods

### 2.1. Study Samples

We conducted individual genome-wide vQTL scans on three cohorts: the Iowa Fluoride Study (IFS, n = 396), the Center for Oral Health Research in Appalachia, cohort 1 (COHRA1, n = 328), and cohort 2 (COHRA2, n = 773). All participants were unrelated children of European ancestry at approximately 3–5 years of age. The study design and recruitment descriptions of the three cohorts have been reported before [7,8,9,10,11]. Briefly, IFS is a longitudinal birth cohort that recruited mothers and newborns from eight Iowa hospital postpartum units between 1992–1995 and has been collecting fluoride exposure, diet, and other information related to dental fluorosis and caries. COHRA1 is a population-based cohort established in 2000 that recruited northern Appalachian families aiming to understand contributing factors to oral diseases. COHRA2 is a longitudinal study and has been recruiting and prospectively following pregnant white women from Pennsylvania and West Virginia and their children starting in 2011. This cohort has information on genetic, microbial, behavioral, and environmental factors involved in oral health. We conducted the power analysis based on the significance level at 1 × 10^−6^, minor allele frequency (MAF) spectrum (0–0.5), and effect size spectrum of phenotypic variability (0.1–1); we had enough power (>80%) to detect associations of SNPs with effect size > 0.1 and MAF > 0.1 in our study sample size. Written parental consent was provided prior to each child’s participation in their respective study and Institutional Review Boards at the University of Iowa, University of Pittsburgh and West Virginia University approved all aspects of the study.

### 2.2. Genotyping

IFS and COHRA1 children were genotyped for about 580,000 SNPs by the Center for Inherited Disease Research at Johns Hopkins University using the Illumina Human610-Quadv1_B BeadChip (Illumina Inc., San Diego, CA, USA). Ungenotyped single nucleotide polymorphisms (SNPs) and sporadic missing data of genotyped autosomal SNPs were imputed for a total of 16.2 million genetic variants using Michigan Imputation Server (Minimac 4) [12] and Haplotype Reference Consortium (HRC) r1.1 as the reference panel [13]. Participants in COHRA2 were genotyped by the same Center using the Infinium Multi-Ethnic Global-8 v1.0 Array and imputed using the Michigan Imputation Server (Minimac 3 for autosomal chromosomes), and HRCr1.1 as the reference panel. Across the three cohorts, variants were excluded if they had low imputation quality (INFO score < 0.3), departed from Hardy-Weinberg equilibrium (*p* < 1 × 10^−6^), or had minor allele frequencies < 10%. Approximately 4.3 million variants were included in the genome-wide vQTL analyses across the three cohorts.

### 2.3. Phenotyping

The decayed and filled surfaces (dfs) index in the primary dentition was obtained through intra-oral examinations in each cohort. We performed the analyses in the subsets of participants after the following processing steps: (1) included children had at least one primary tooth; (2) biological relatives were excluded (with one member per kinship retained at random); (3) genetic ancestry outliers were excluded based on calculated principal components (PCs) of ancestry using principal component analysis (PCA) of genome-wide genetic data and excluding participants with PC1 and PC2 more than 3 standard deviations (SDs) from the mean; (4) phenotype outliers were excluded by adjusting the raw phenotype (i.e., dfs) for age, number of surfaces, and the first 6 PCs of ancestry and excluding participants with adjusted phenotypes more than 3 SDs from the mean. The number of adjusted PCs sufficient to capture the population structure was determined based on the scatterplots and scree plots of PCs; (5) the adjusted phenotype was standardized to z scores with mean 0 and variance 1 in each sex group. Note, the phenotype processing steps not only removed the effects of age, number of surfaces and the first 6 PCs on the phenotype, but also the differences in mean and variance between the two sex groups. The workflow of this process and the number of excluded participants in each step are summarized in Figure 1.

### 2.4. Genome-Wide vQTL Scans and Meta-Analysis

We used Levene’s test with the median as implemented in OSCA software [14] to test the associations between genetic variants with the variance of dfs in the three cohorts separately. After completing the individual genome-wide vQTL analyses in the three cohorts, we conducted Stouffer’s *p*-value-based meta-analysis using the summary statistics from the three cohorts via METAL [15] in order to increase statistical power. We set a threshold for genome-wide significance at *p* < 5 × 10^−8^ and for suggestive significance *p* < 1 × 10^−6^. The results were summarized and visualized using Manhattan plots and quantile-quantile (Q-Q) plots created in R. We prioritized vQTLs with *p* < 1 × 10^−6^, which were generated from three cohorts and meta-analysis of three cohorts, then used these prioritized vQTL in the following GEI analysis.

### 2.5. Environmental Factors

In the following GEI analysis, we tested the interaction effects on dental caries in the primary dentition separately in the three cohorts using prioritized variants and environmental factors. For the environmental factors, we included available demographic, behavioral, and environmental factors that putatively relate to oral health across the three cohorts. The tested environmental factors were different across three cohorts.

There were 14 factors tested in IFS including sex (male or female), annual household income (<$20,000 or $20,000–39,999 or ≥$40,000), mother’s educational attainment (up to high school or some college or above four-year degree), father’s educational attainment (up to high school or some college or above four-year degree), water source (city/public or well), home water fluoride level (ppm, measured in a water sample), daily brushing frequency, birth weight (kg), gestational weeks at birth, daily milk intake (oz), daily 100% juice intake (oz), daily sugar-sweetened beverage (SSB) intake (oz), daily fluoride intake (mg), and daily powdered beverage intake (oz). In COHRA1, ten factors were tested including sex (male or female), recruitment site (Pennsylvania [PA] or West Virginia [WV]), annual mother’s income (<$15,000 or ≥$15,000), annual father’s income (<$25,000 or ≥$25,000), mother’s educational attainment (≤high school or >high school), father’s educational attainment (≤high school or >high school), water source (city/public or well), home water fluoride level (ppm), daily brushing frequency, and fluoride supplement (yes or no). In COHRA2, ten factors were tested including sex (male or female), recruitment site (PA or WV), annual mother’s income (<$25,000 or ≥$25,000), home water source (city/public or well), home water fluoride level (ppm), mother’s educational attainment (<high school, high school or some college, or college degree and more), mother’s tooth brushing frequency (>1/day, 1/day, or <1/day), breastfeeding history (yes or no), breastfeeding duration (months), and sugar-sweetened beverages intake (SSB, yes or no). We calculated the correlation coefficients (r) among the tested factors in the three cohorts separately and did not find any strong correlations (defined as r > 0.8).

### 2.6. Genotype-by-Environment Interaction Analysis

We used linear regression to test the interactions between the prioritized vQTLs and each of the environmental factors in the three cohorts separately. Environmental factors in each cohort were described above. The model for GEI analysis was:$$y=\mu +\beta_{G}x_{G}+\beta_{E}x_{E}+\beta_{GE}x_{G}x_{E}+e$$
where y is the standardized phenotype, μ is the mean term, $x_{G}$ is the mean-centered SNP genotype, and $x_{E}$ is the mean-centered environmental factor. The βs are the coefficients for the genotype, the environmental factor, and the GEI, respectively. A stringent Bonferroni correction (i.e., 0.05 divided by the number of SNPs and the number of factors tested for GEI) was used to determine the significance thresholds for GEI effects, which were 9.16 × 10^−5^ (i.e., 0.05/39 SNPs/14 factors) for IFS, 1.28 × 10^−4^ (i.e., 0.05/39 SNPs/10 factors) for COHRA1 and 1.28 × 10^−4^ (i.e., 0.05/39 SNPs/10 factors) for COHRA2.

## 3. Results

### 3.1. Sample Summary

The summary of environmental factors in the three cohorts is listed in Table 1. After applying quality filters for the participants, a total of 1497 children across the three cohorts were included in this analysis. There were six overlapping environmental factors across three cohorts including sex, parental education information, income information, home fluoride level, tooth brushing frequency, and water source. Other factors were cohort-specific.

### 3.2. Genome-Wide vQTL Scans and Meta-Analysis

We conducted separate genome-wide vQTL scans of dfs in IFS, COHRA1, and COHRA2 using Levene’s test and combined results across the three cohorts using meta-analysis. The Manhattan plots for the vQTL analysis of each cohort and the meta-analysis are shown in Figure 2. The genomic inflation factors were 1.00, 1.02, and 1.03 in IFS, COHRA1, and COHRA2, respectively, indicating there was no genomic inflation (Appendix A shows the Q-Q plots for vQTL scans). There were three loci reaching the genome-wide significance threshold (*p* < 5 × 10^−8^) in IFS, two loci in COHRA1, three loci in COHRA2, and two loci in the meta-analysis. In addition, there were eight signals at suggestive level (5 × 10^−8^ < *p* < 1 × 10^−6^) in IFS, seven signals in COHRA1, 12 signals in COHRA2, and three signals in meta-analysis. In total there were 40 signals, 10 at the genome-wide significance threshold and 30 at the suggestive significance threshold. Notably, rs2090166 (the top SNP from COHRA2) and rs3862191 (from the meta-analysis) were in high linkage disequilibrium (r^2^ = 0.98). Therefore, a total of 39 independent SNPs were used in the following GEI analysis. Table 2 lists results for the 40 top SNPs from the three cohorts and meta-analysis.

### 3.3. Genotype-by-Environment Interaction Analysis

There were three significant GEI effects in IFS: rs9830884 with household income (*p* = 4.24 × 10^−5^), rs1491071 with father’s educational attainment (*p* = 4.25 × 10^−6^), and rs1978471 with tooth brushing frequency (*p* = 3.15 × 10^−5^) (results for all 39 GEI in IFS are listed in Appendix A). Figure 3 shows the form of the interactions between the SNPs and environmental factors. Children with the CC genotype of rs1491071 had higher standardized dfs than the CT and TT genotypes when their father’s educational attainment was less than four years of college. Similarly, children with the TT genotype of rs1967471 exhibited higher dfs than the CT and CC genotypes when their tooth brushing frequency was less than two times a day and children with the CC genotype of rs9830883 had higher caries compared to the AC and CC genotypes when their household income was less than $40,000/year (Figure 3a–c).

There was one significant GEI in COHRA1, rs7463853 with father’s income (*p* = 3.50 × 10^−5^) (results for all 39 GEI in COHRA1 were listed in Appendix A). Children with the GG genotype of rs7463853 had higher caries experience in the primary dentition compared to those with the AA and AG genotypes when their father’s income was less than $25,000/year (Figure 3d).

There were four significant GEI effects in COHRA2 including rs71508615 with sex (*p* = 6.78 × 10^−6^) and mother’s income (*p* = 1.25 × 10^−4^), rs9685188 with breastfeeding status (*p* = 8.03 × 10^−5^), and rs73723358 with SSB (*p* = 7.39 × 10^−5^) (results for all GEI in COHRA2 were listed in Appendix A). Children with the GG genotype of rs71508615 exhibited higher standardized dfs than the AA and AG genotypes when their sex was female, and their mother’s income was less than $25,000/year. Similarly, children with the TT genotype of rs9685188 had higher standardized dfs than the CT and CC genotypes when they were not breastfed and children with the GG genotype of rs73723358 experienced higher caries experience than the AG and AA genotypes when they had greater consumption of sugar-sweetened beverages (Figure 3e–h).

## 4. Discussion

This study explored genetic effects on the variance of dental caries experience and presented the application of genome-wide vQTL scans for prioritizing SNPs for GEI detection. The fact that the loci in this study were detected through vQTL scans (but not tests of the trait mean) indicates the unique value of studies focusing on phenotypic variance for studies of complex traits like dental caries. This is supported by the fact that only one of our discovered loci (*NAMPT*) was previously reported to be associated with caries in the permanent dentition. In addition, by using genome-wide vQTL scans to prioritize SNPs, we identified several significant GEI effects.

The genome-wide vQTL scans of dental caries in the three separate cohorts and the meta-analysis of the three cohorts cumulatively identified 39 independent loci that were significantly associated with caries variance. The top SNPs from the 39 identified vQTL signals were used to test the GEI effects on dental caries in the primary dentition, yielding eight significant GEI in total across the three cohorts. Two of the SNPs from the eight significant GEI are in or near the genes that were reported to be involved in dental caries with relatively strong evidence in previous studies. First, the SNP rs73723358 identified as a possible vQTL at the suggestive level in COHRA2 is located near the gene *NAMPT* that was previously reported to be associated with dental caries in the permanent dentition [16] in a Hispanic/Latino population. *NAMPT* is a pro-inflammatory adipokine that is expressed in periodontal tissue [17] and is involved in periodontal healing and regulating matrix-degrading enzymes [18]. Though the direct role of *NAMPT* in dental caries is not clear and rs73723358 does not have a known eQTL or regulatory effects on *NAMPT*, this SNP interacted with SSB intake influencing caries in the primary dentition in our study. Sugar in food and drinks is a well-known risk factor for dental caries [19], which is used by bacteria as energy and increases the level of produced acids leading to a lower pH in the dental biofilm and demineralization. Children with the GG genotype of rs73723358 had higher caries experience compared to AG and AA only if they had higher SSB consumption, suggesting that the effect of the genetic variant rs73723358 on caries experience in the primary dentition was moderated by the consumption of SSB (and vice versa).

Second, the SNP rs9685188 was identified as a vQTL from the meta-analysis of the three cohorts at a genome-wide significance level and is located near the gene *IGFBP7*, which encodes a member of the insulin-like growth factor-binding protein family. *IGFBP7* was found to be expressed in the whole cytoplasm of odontoblasts and the intercellular space of maturation-stage ameloblasts during tooth germ mineralization. Knock-down of *IGFBP7* expression in rats promoted dentin matrix mineralization suggesting the negative regulation of *IGFBP7* in enamel and dentin formation [20]. Given the role of *IGFBP7* in tooth development and mineralization, it is plausible that *IGFBP7* impacts caries development through abnormal mineralization. In addition, this SNP influenced caries by significantly interacting with breastfeeding status in our study. Children with the TT genotype of rs9685188 had higher caries scores compared to CT and CC only if they were not breastfed. Breastfeeding is considered a protective factor for dental caries as the milk is released into the throat via breastfeeding, as opposed to bottle feeding, during which the milk is delivered into the front of the mouth and around the teeth. Our results suggest that the effects of *IGFBP7* on dental caries, potentially through its role in tooth development and mineralization, may be different between breastfed and non-breastfed children. However, the direct relationship between caries and this GEI is not clear yet.

Our findings of the effects of GEI on dental caries provide additional insights into caries etiology. First, we identified vQTL signals that might represent novel caries susceptibility loci with effects that were not able to be detected by testing mean effects. Second, the nominated genes might play roles in the biology of dental caries, expanding the list of putatively caries-associated genes for further investigation. Third, the environmental factors interacting with genetic variants in dental caries expand our knowledge of cariogenesis and could potentially help with identifying genetic subgroups with higher exposure-specific caries risk for prevention efforts (i.e., increased environmental-specific risk for individuals with a certain genotype). For instance, socioeconomic status (SES) might influence caries risk through living conditions, behavior, and dietary patterns. Here, we also detected that SES indicators (e.g., income, education) significantly interacted with some of the genetic variants across all three cohorts. Our results indicated that people with lower SES might be more susceptible to dental caries when they carry a certain genotype. A similar scenario may play out for two other risk factors, tooth brushing frequency, and sex; people who brush their teeth less frequently and females experience higher caries risk when they have certain genotypes. Toothbrushing frequency has been reported to be associated with caries incidence and increment of carious lesions [21]. However, we only recorded the tooth brushing frequency without the fluoride information on the toothpaste, so we cannot separate the effectiveness of fluoride toothpaste from toothbrushing frequency. Thus, the detailed mechanisms of these identified GEI in dental caries need to be further explored.

In addition to the findings on GEI effects for dental caries, the vQTL scans, themselves, identified several signals located in or near genes with plausible biological roles in dental caries. For example, in the IFS cohort, the top SNP at the genome-wide significance level, rs59190052, has eQTL effects on nearby genes *ANO4* and *UTP20* and is located near the gene *SLC5A8*, which was reported to be expressed in ameloblasts at the maturation stage and directly involved in enamel mineralization [22]. Abnormal *SLC5A8* expression disrupted ion and pH homeostasis and inhibited enamel crystal growth, which led to enamel hypomineralization and might contribute to the development of dental caries. In addition, *SLC5A8* has been shown to be expressed in epithelial cells and stimulate apoptosis, thus it may also promote apoptosis and autophagy of ameloblast during amelogenesis [23]. Likewise, in COHRA1, the vQTL SNP rs11970843, which was identified at the suggestive level is located near the gene *SHH* and has putative regulatory effects on the enhancer of *SHH*, which encodes the Sonic Hedgehog signaling molecule. The Shh signaling pathway plays an essential role in human tooth development from initiation through to root development, which includes setting the boundaries between odontogenic and non-odontogenic epithelium at the initiation stage, and regulates cell cycle, differentiation, and morphogenesis at the stage of tooth germ establishment. Moreover, recent studies have shown that the Shh signaling pathway promotes morphogenetic movement via cell polarization, impacts both cuspal and root development at later stages, and defines tooth number at the maturation stage [24]. However, the mechanisms of *SHH* in caries development are not clear and need to be investigated in the future. Another possible vQTL observed at the suggestive level in COHRA2, rs622516 is in the gene *PKD2L1* and has eQTL effects on *PKD2L1* in the testis, frontal cortex, and cortex tissues. *PKD2L1* encodes a member of the transient receptor potential family of ion channels and is a potential candidate gene for the sour taste that detects acids in foods and drinks, which may influence dietary patterns and potentially influence dental caries. Knockout *PKD2L1* in a mouse model reduces the responses of the gustatory nerve to sour taste stimuli [25].

There are several limitations of this study. The sample sizes are relatively small in the three cohorts, and even in the meta-analysis. SNPs with small effect sizes on dental caries variance might not have been detected due to low power. There were no overlapping vQTLs among the three cohorts (i.e., the identified loci in one cohort were not replicated in the other two cohorts even at nominal *p*-values), even with the same phenotype and similarly aged participants. This may also be a consequence of the low power problem. Another explanation is that the vQTL effects and GEI are cohort-specific. Participants were recruited from different regions in different decades and were exposed to different levels of environmental factors. Thus, the identified vQTL in one population may not be generalizable to other populations. Another limitation is that variance heterogeneity can be explained by multiple mechanisms including genuine interaction effects, confounding by phenotypic distribution, and induction by nearby causal variants. We are not able to fully explore all the potential mechanisms to inform our interpretations of the observed vQTLs. After prioritizing the SNPs via the vQTL scans, we only explored the GEI effects on dental caries. Genotype-by-genotype interactions can also lead to vQTLs, which is out of the scope of this study but is worth examining in the future. Another potential limitation is that for some of the vQTLs, we may not have collected data on the truly interacting environmental factor, hence, we were not able to detect the corresponding GEI for those vQTLs. Similarly, there were missing values for some of the environmental factors in COHRA1 (e.g., mother’s and father’s educational attainment) and an unbalanced number of participants in the strata of factors and genotypes in COHRA1 and COHRA2, which might cause problems in detecting the accurate effects of GEI on dental caries and replicating the GEI effects in other cohorts.

## 5. Conclusions

We conducted the first genome-wide vQTL scan of dental caries and identified 39 independent loci that were associated with caries variance at significant or suggestive levels. Some of the identified loci are in or near genes with possible functions in dental caries. Using these prioritized SNPs, we detected several environmental factors that significantly interacted with the vQTL SNPs to influence dental caries experience, including income, education, tooth brushing frequency, sex, breastfeeding, and SSB consumption. Our results expand the understanding of dental caries etiology.

## Figures and Tables

**Figure 1 genes-14-00736-f001:**
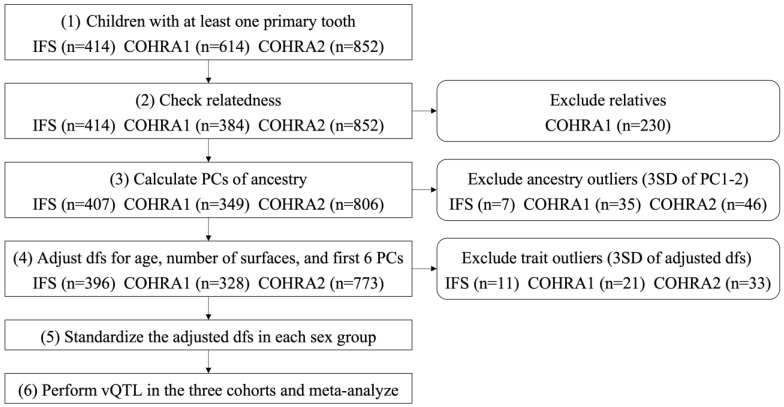
Participant process workflow.

**Figure 2 genes-14-00736-f002:**
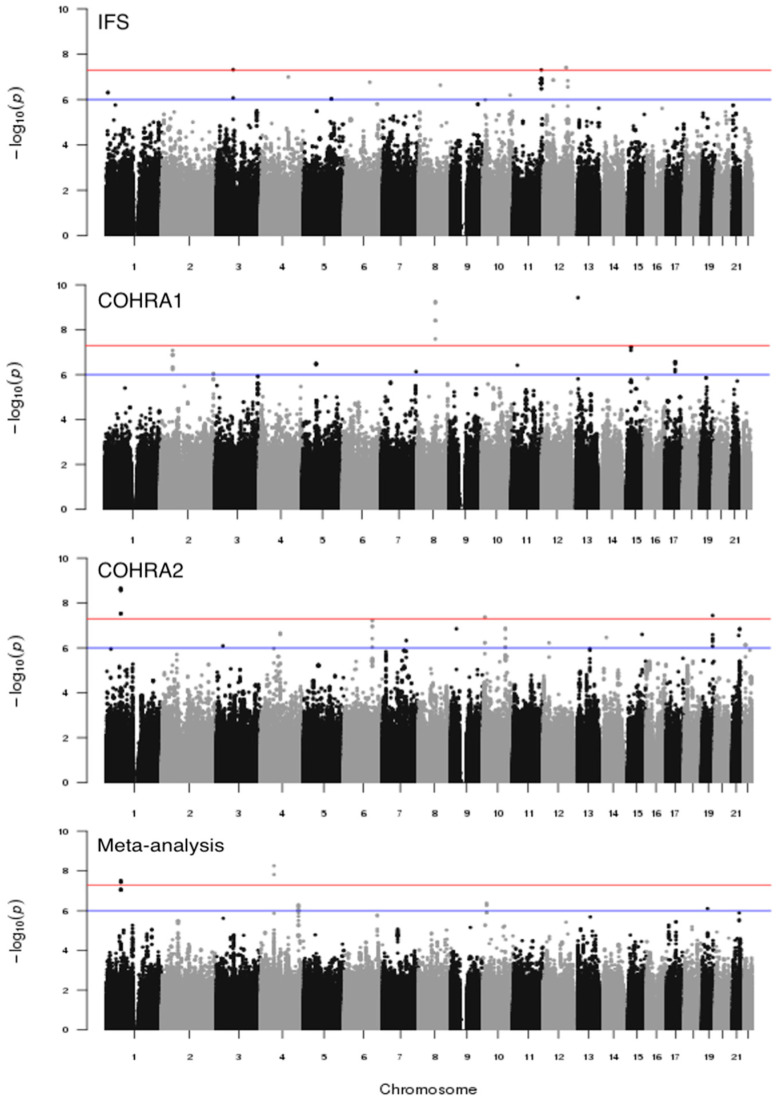
Manhattan plot for genome-wide vQTL scans of dfs in IFS, COHRA1, COHRA2, and meta-analysis; −log_10_-transformed *p*-values (y-axis) for each SNP organized by physical position across the chromosomes (x-axis) are shown. Horizontal lines represent the genome-wide significant (5 × 10^−8^) and suggestive (1 × 10^−6^) *p*-value thresholds.

**Figure 3 genes-14-00736-f003:**
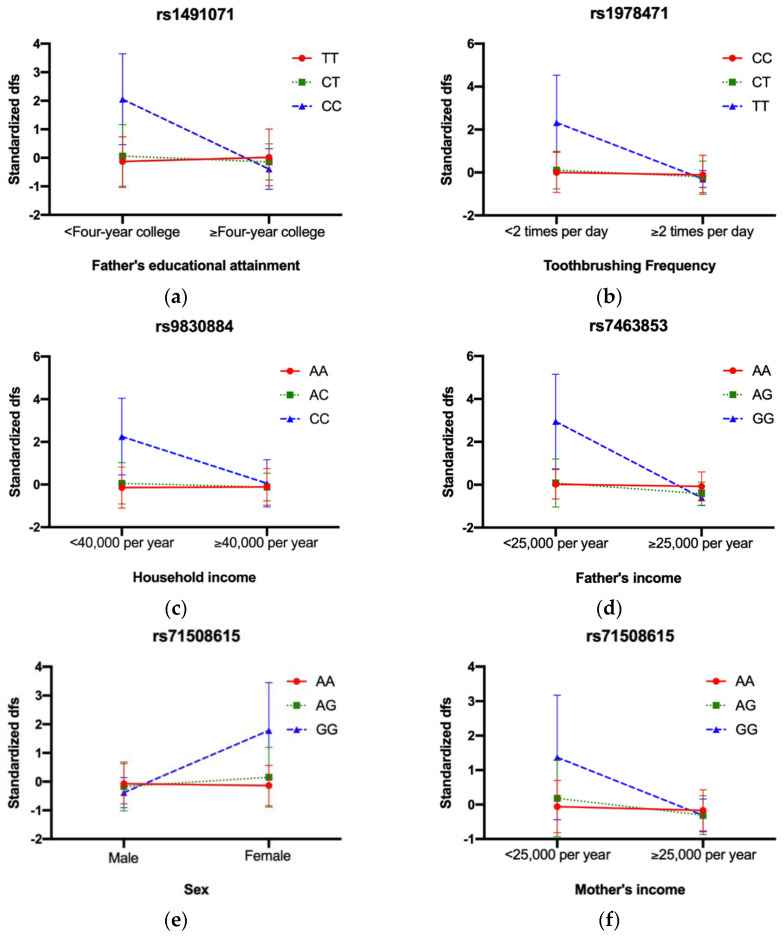
Interaction plots for significant GEI across three cohorts. In IFS, (**a**) rs1491071 with father’s educational attainment, CC genotype had higher dfs when father’s educational attainment < Four−year college; (**b**) rs1978471 with toothbrushing frequency, TT genotype had higher dfs when toothbrushing frequency < 2 times per day; (**c**) rs9830884 with household income, CC genotype had higher dfs when household income < 40,000 per year; In COHRA1, (**d**) rs7463853 with father’s income, GG genotype had higher dfs when father’s income < 25,000 per year; In COHRA2, (**e**) rs71508615 with sex, GG genotype had higher dfs when sex was female; (**f**) rs71508615 with mother’s income, GG genotype had higher dfs when mother’s income < 25,000 per year; (**g**) rs9685188 with breastfeeding status, TT genotype had higher dfs when breastfed; (**h**) rs73723358 with SSB, GG genotype had higher dfs when intaking SSB.

**Table 1 genes-14-00736-t001:** Basic characteristics of study samples in IFS, COHRA1, and COHRA2.

Variables	IFS (n = 396)	COHRA1 (n = 328)	COHRA2 (n = 773)
Decayed and filled surfaces (dfs)	1.36 ± 2.81	1.47 ± 3.22	0.68 ± 2.04
Age at examination	5.14 ± 0.41	3.53 ± 1.52	3.69 ± 1.40
Sex (Male vs. Female)	189/207	166/162	405/368
Site (PA vs. WV)	NA	94/234	437/336
Mother’s educational attainment	<4-year college: 215 ≥4-year college: 181	≤high school: 133 >high school: 113	<4-year college: 336 ≥4-year college: 434
Father’s educational attainment	<4-year college: 214 ≥4-year college: 169	≤high school: 90 >high school: 59	NA
Mother’s income	NA	<$15,000: 108 ≥$15,000: 68	<$25,000: 422 ≥$25,000: 339
Father’s income	NA	<$25,000: 62 ≥$25,000: 72	NA
Household Income	<$40,000: 185 ≥$40,000: 196	NA	NA
Home fluoride level (ppm)	0.80 ± 0.40	0.66 ± 0.43	0.78 ± 0.29
Brushing frequency (≥2/day vs. <2/day)	205/175	176/124	594/76
Water source (city/public vs. well)	297/85	249/62	661/40
Birth weight (kg)	3.50 ± 0.58	NA	NA
Gestational weeks	39.40 ± 2.10	NA	NA
Daily milk intake (oz)	13.00 ± 6.10	NA	NA
Daily 100% juice intake (oz)	5.72 ± 4.17	NA	NA
Daily SSB intake (oz)	4.83 ± 3.92	NA	NA
Daily fluoride intake (oz)	0.71 ± 0.35	NA	NA
Daily powdered beverage intake (oz)	1.37 ± 3.70	NA	NA
Breastfeeding duration (months)	NA	NA	3.92 ± 5.61
Breastfeeding status (Yes vs. No)	NA	NA	643/106
SSB intake	NA	NA	No: 460 Yes: 313

Note: IFS: Iowa Fluoride Study; COHRA1: Center for Oral Health Research in Appalachia, cohort 1; COHRA2: Center for Oral Health Research in Appalachia, cohort 2; PA: Pennsylvania; WV: West Virginia. SSB: sugar-sweetened beverages. For categorical variables, showing the number of participants in each group; for numeric variables, showing the mean ± standard deviation. NA: not available in the cohort.

**Table 2 genes-14-00736-t002:** 40 prioritized SNPs (*p* < 1 × 10^−6^) from vQTL of dfs in IFS, COHRA1, COHRA2, and meta-analysis.

No	SNP	CHR	POS	A1	A2	Frequency	β	SE	*p*	N	Cohort
1	rs59190052	12	101640969	A	G	0.23	0.447	0.08	3.92 × 10^−8^	396	IFS
2	rs9830884	3	73612432	C	A	0.29	0.411	0.08	4.82 × 10^−8^	396	IFS
3	rs77322490	11	125701344	T	C	0.19	0.48	0.09	4.92 × 10^−8^	396	IFS
4	rs6844159	4	121794535	C	T	0.3	0.4	0.08	1.01 × 10^−7^	396	IFS
5	rs3947271	12	43735470	A	C	0.19	0.458	0.09	1.39 × 10^−7^	396	IFS
6	rs1089941	12	108759443	T	G	0.16	0.491	0.09	1.47 × 10^−7^	396	IFS
7	rs1491071	6	113566803	C	T	0.22	0.436	0.08	1.73 × 10^−7^	396	IFS
8	rs2018981	8	98266984	T	A	0.13	0.535	0.1	2.32 × 10^−7^	396	IFS
9	rs11587481	1	7611159	G	T	0.19	0.443	0.09	4.93 × 10^−7^	396	IFS
10	rs11199332	10	122186330	A	G	0.11	0.557	0.11	6.38 × 10^−7^	396	IFS
11	rs11241707	5	123005420	A	G	0.45	−0.34	0.07	9.34 × 10^−7^	396	IFS
12	rs12429729	13	27568586	A	C	0.1	0.769	0.12	3.68 × 10^−10^	325	COHRA1
13	rs7463853	8	82662694	G	A	0.23	0.547	0.09	5.48 × 10^−10^	326	COHRA1
14	rs690435	15	41132310	C	T	0.31	0.438	0.08	5.78 × 10^−8^	328	COHRA1
15	rs12994450	2	53649620	C	T	0.29	0.44	0.08	8.26 × 10^−8^	328	COHRA1
16	rs11654217	17	45531884	G	T	0.23	0.462	0.09	2.73 × 10^−7^	328	COHRA1
17	rs264532	5	61624327	C	T	0.14	−0.55	0.11	3.33 × 10^−7^	328	COHRA1
18	rs12797571	11	25880475	G	C	0.51	0.382	0.08	3.80 × 10^−7^	328	COHRA1
19	rs11970843	7	155917252	A	G	0.21	0.497	0.1	7.33 × 10^−7^	278	COHRA1
20	rs4663531	2	235893092	G	A	0.25	0.429	0.09	8.73 × 10^−7^	328	COHRA1
21	**rs2090166**	1	64750226	C	T	0.14	0.434	0.07	2.33 × 10^−9^	773	COHRA2
22	rs3786738	19	48716187	T	C	0.1	0.458	0.08	3.59 × 10^−8^	772	COHRA2
23	rs11817228	10	10340951	C	T	0.11	0.44	0.08	4.22 × 10^−8^	771	COHRA2
24	rs512158	6	125415660	G	A	0.12	0.424	0.08	5.82 × 10^−8^	769	COHRA2
25	rs622516	10	102082135	C	T	0.15	0.374	0.07	1.31 × 10^−7^	760	COHRA2
26	rs71508615	9	24787071	G	A	0.11	0.423	0.08	1.40 × 10^−7^	772	COHRA2
27	rs9982623	21	47691216	T	C	0.13	0.392	0.07	1.44 × 10^−7^	773	COHRA2
28	rs2869342	4	86320413	T	C	0.31	0.281	0.05	2.20 × 10^−7^	768	COHRA2
29	rs17536922	15	85461106	T	G	0.11	0.418	0.08	2.50 × 10^−7^	728	COHRA2
30	rs10651815	21	43045398	G	C	0.27	−0.29	0.06	2.77 × 10^−7^	759	COHRA2
31	rs1958016	14	33605479	C	A	0.19	0.338	0.07	3.42 × 10^−7^	713	COHRA2
32	rs73723358	7	106399401	G	A	0.11	0.403	0.08	4.68 × 10^−7^	752	COHRA2
33	rs7972868	12	26695988	C	T	0.18	0.326	0.07	5.96 × 10^−7^	769	COHRA2
34	rs73157913	22	25284280	G	T	0.12	0.384	0.08	7.39 × 10^−7^	740	COHRA2
35	rs11923408	3	28647903	C	G	0.28	0.282	0.06	8.14 × 10^−7^	745	COHRA2
36	rs9685188	4	58066166	T	C	0.31	5.829	NA	5.57 × 10^−9^	1446	Meta
37	**rs3862191**	1	64753505	T	C	0.17	5.547	NA	2.90 × 10^−8^	1486	Meta
38	rs11592458	10	17053179	C	G	0.12	5.064	NA	4.10 × 10^−7^	1487	Meta
39	rs1497945	4	167043469	A	T	0.35	5.03	NA	4.91 × 10^−7^	1496	Meta
40	rs1978471	19	24024926	T	C	0.18	4.942	NA	7.74 × 10^−7^	1489	Meta

Note: CHR: chromosome; POS: position; A1: effect allele; A2: reference allele; Frequency: frequency of effect allele; β: effect size of effect allele; SE: standard error; *p*: *p*-values; N: sample size; IFS: Iowa Fluoride Study; COHRA1: Center for Oral Health Research in Appalachia, cohort 1; COHRA2: Center for Oral Health Research in Appalachia, cohort 2. rs2090166 and rs3862191 (bolded in SNP column) are in high linkage disequilibrium with r^2^ = 0.98. The significant level was *p* < 1 × 10^−6^.

## Data Availability

The data presented in this study are available on request from the corresponding author. The data are not publicly available due to privacy.

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
