# Peer review of "Genome-Wide Analysis of Dental Caries Variability Reveals Genotype-by-Environment Interactions"

_genes, 2023, doi:10.3390/genes14030736_

Round 1
Reviewer 1 Report
This is a substantial study on an important topic. However, before recommending its acceptance there are a number of points to be addressed by the authors.
The Introduction, Materials and Methods, and Results are clear and well written and the study design is appropriate.
The points to be addressed are:-
- 'Tooth brushing frequency', Line 318- there are clinical studies that have shown that tooth brushing itself does not reduce caries incidence. Rather it is the use of a fluoride containing toothpaste that is effective. While the data originally gathered may only have recorded brushing frequency, a short comment in the Discussion making this point is needed, together with an appropriate reference.
- The statement, Line 329 - 'Enamel hypomineralisation, thus increasing susceptibility to caries development' requires references to substantiate this point. Patients with marked enamel hypomineralisation in some types of Amelogenesis Imperfecta are caries free.
- The statement, Line 342 - 'It is possible that SHH influences dental caries through abnormal development', similarly needs clarification and justification. What is the evidence that SHH affects enamel development and how does this increase susceptibility to caries?
- The References List- 10 of the 25 references do not include the full details of Authors, Journals, publication dates and volume and page numbers. It is not clear why 15 references are presented in full and the other 10 are not.
- Two minor grammatical suggestions:-
- Line 34 . Insert 'factors' after 'genetic'
- line 92 . Remove 'that' fromstart of line.
Reviewer 2 Report
1. Write in the first person 2. Extend the conclusions
Reviewer 3 Report
Title
Appropriate
Abstract
The abstract should follow the style of structured abstracts, but without headings.
Line 18: Why “White” children only
All abbreviations need to be in full term first.
In the method part, please add method of statistical analysis. Also, add real P-values.
Line 27: Please correct “report” to “reported”.
Introduction
English language editing is required.
The Introduction does not give a rationale why this study should be conducted or what scientific value it has.
At the end of the Intro section, please give your null hypothesis. The latter should be derived from the preceding thoughts in this section and should be broached again in the Discussion. In hypothesis testing, the null hypothesis is the one you are hoping that is can be disproven by the observed data.
Materials and Methods
What is the study rationale?
Please mention sample size calculation.
Please mention study design.
Please add subtitle inclusion and exclusion criteria.
Was the study registered in ClinicalTrials.gov
Please add abbreviation following full term when possible. The first time an abbreviation appears, it should be placed in parentheses following the full spelling of the term.
Please add the name of the manufacturer for all materials and equipment used.
Please add the name of the manufacturer for the statistical software used.
Results
English language editing is required.
Please add real P-values if they were calculated.
Tables and Figures
In Tables 1 and 2, a footnote explaining the abbreviations (IFS, P, N etc..) needs to be added (what do they stands for). Also, level of significance needs to be mentioned.
Discussion
This section may usefully start with a summary of the major findings, but repetition of parts of the abstract or of the results section should be avoided.
Please mention the null hypothesis if accepted or rejected.
Please mention future directions.
Please point out the implications of the findings and their limitations.
Conclusions:
Lengthy.
References
Also, please check journal guidelines for reference writing.
References needs to be 10 years back not more (from 2012 to 2022).
Old references need to be replaced by recent ones.
All of the journal names need to be abbreviated.
Some of the references include DOI, others do not include DOI number.
In general, all references need to be revised, standardized and written according to the journal guidelines.

Round 2
Reviewer 3 Report
None. Thank you for your replies.